# The Metabolic Profile of *Anchusa officinalis* L. Differs According to Its Associated Arbuscular Mycorrhizal Fungi

**DOI:** 10.3390/metabo12070573

**Published:** 2022-06-22

**Authors:** Evangelia Tsiokanos, Annalisa Cartabia, Nikolaos Tsafantakis, Ismahen Lalaymia, Aikaterini Termentzi, Maria Miguel, Stéphane Declerck, Nikolas Fokialakis

**Affiliations:** 1Department of Pharmacy, Division of Pharmacognosy and Natural Product Chemistry, National and Kapodistrian University of Athens, 15771 Athens, Greece; evatsiokanos@pharm.uoa.gr (E.T.); ntsafantakis@pharm.uoa.gr (N.T.); 2Applied Microbiology, Mycology, Earth and Life Institute, Université Catholique de Louvain-la-Neuve, 1348 Louvain-la-Neuve, Belgium; annalisa.cartabia@uclouvain.be (A.C.); ismahen.lalaymia@uclouvain.be (I.L.); stephan.declerck@uclouvain.be (S.D.); 3Laboratory of Pesticides’ Toxicolology, Benaki Phytopathological Institute, 8 St. Delta Street Kifissia, 14561 Athens, Greece; a.termentzi@bpi.gr; 4Instituto de Tecnologia Química e Biológica António Xavier, Universidade Nova de Lisboa (ITQB NOVA), Av. da República, 2780-157 Oeiras, Portugal; pmariamiguel@gmail.com

**Keywords:** arbuscular mycorrhizal fungi, *Anchusa officinalis* (L.), circulatory semi-hydroponic cultivation system, metabolomics, primary and secondary metabolites

## Abstract

*Anchusa officinalis* (L.) interacts with various microorganisms including arbuscular mycorrhizal fungi (AMF). Recently, the AMF *Rhizophagus irregularis* MUCL 41833 has been shown to modulate the metabolome of *A. officinalis*. However, little information is available on the impact that different AMF species may have on primary and secondary plant metabolites. In this study, four AMF species belonging to the genus *Rhizophagus* (*R. irregularis* MUCL 41833, *R. intraradices* MUCL 49410, *R. clarus* MUCL 46238, *R. aggregatus* MUCL 49408), were evaluated for their potential to modulate *A. officinalis* metabolome under controlled semi-hydroponic cultivation conditions. An untargeted metabolomic analysis was performed using UHPLC-HRMS followed by a multivariate data analysis. Forty-two compounds were reported to be highly modulated in relation to the different AMF associations. Among them, six new secondary metabolites were tentatively identified including two acetyl- and four malonyl- phenylpropanoid and saponin derivatives, all presenting a common substitution at position C-6 of the glycosidic moiety. In addition, an enhanced accumulation of primary and secondary metabolites was observed for *R. irregularis* and *R. intraradices*, showing a stronger effect on *A. officinalis* metabolome compared to *R. clarus* and *R. aggregatus*. Therefore, our data suggest that different AMF species may specifically modulate *A. officinalis* metabolite production.

## 1. Introduction

The interactions between plants and beneficial microorganisms improve not only growth and health, but also modify the metabolome considerably [1,2]. Among these microorganisms, arbuscular mycorrhizal fungi (AMF) are known for their intimate symbiotic relationship with circa 72% of land plants [3]. They facilitate phosphate and nitrogen uptake in exchange for carbohydrates [4,5], offer enhanced resistance to pests and diseases, drought and salinity, and are involved in the heavy metal detoxification process [6].

An increasing number of studies (reviewed extensively by Zeng et al. [7] and by Kaur and Suseela [2]) have reported the impact of AMF on the production of primary and secondary metabolites in different plant species, including crops. For instance, a higher accumulation of amino acids, fatty acids, isoflavonoids and phenylpropanoid derivatives was reported in roots of *Medicago truncatula* colonized by *Rhizophagus irregularis* [8]. Similarly, a significant upregulation of secondary metabolites belonging mainly to blumenol derivates and pyrrolizidine alkaloids, was detected in the roots of *Senecio jacobaea* associated with *R. irregularis* [9].

Not necessarily all plant-AMF associations result in increased metabolite production. Τhe productivity of the host plant—i.e., the outcome of the plant-microbe interaction—is highly dependent on the identity of the AMF symbiont [2]. Indeed, different AMF species can differently affect the production of specific phytochemicals on a specific plant species. For instance, Rivero et al. [10] reported that, although the metabolic pathways of *Solanum lycopersicum* altered by mycorrhizal symbiosis were common for both *Funneliformis*
*mosseae* and *R. irregularis*, the compound’s biosynthesis was altered depending on the particular AMF species involved. Jasmonic acid (JA), methyl-JA and JA-Isoleucine conjugates were accumulated in significantly higher amounts only in the plants associated with *F. mosseae*. This suggests that specific AMFs have a fine-tuned regulation role in biosynthetic pathways.

Besides the increasing interest in plant-AMF associations, those studies applying a metabolomic approach were limited to only two AMF species, *R. irregularis* (formerly *Glomus intraradices*) and *F. mosseae* (formerly *Glomus mosseae*) [8,9,10,11,12,13]. In a recent study, Cartabia et al. [14] described the effects of *R. irregularis* MUCL 41833 on shoots and roots of *A. officinalis*, growing under a semi-hydroponic cultivation system for a period of 9 days. The untargeted metabolomic approach showed an upregulation of 35 primary and secondary metabolites (e.g., organic acids, phenolic compounds, oleanane-types glycosides) in mycorrhized plants as compared to the non-mycorrhized ones. Interestingly, *A. officinalis* is always found associated with AMF in the wild [15]. Thus, considering non-mycorrhized plants as control does not truly reflect the natural conditions. Therefore, in the present study, a metabolomic analysis of roots and shoots of *A. officinalis* associated with four different AMF species belonging to the same genus (*R. irregularis* MUCL 41833, *R. intraradices* MUCL 49410, *R. clarus* MUCL 46238 and *R. aggregatus* MUCL 49408) was conducted to test the hypothesis that different AMF strains impact the plant metabolome differently. *R. irregularis* MUCL 41833 was chosen as the reference strain since it is considered as a model organism in AMF research and the most used AMF strain in commercial inoculants [10,16]. It is a generalist colonizer, present in almost all soils and climatic zones [16], readily grown in vitro on root organs [17] and its whole genome sequence has been published recently [18,19]. Moreover, its effects on the metabolome of *A. officinalis* were recently published [14].

To test our hypothesis, the plants were grown for 9 days in the semi-hydroponic cultivation system described by Cartabia et al. [14] and an ultrahigh-performance liquid chromatography high-resolution mass spectrometry (UHPLC-HRMS) analysis was performed. An untargeted metabolomics approach was further conducted to shed light on the overall effects of the different AMF species on *A. officinalis* root and shoot parts.

## 2. Results

### 2.1. Root Colonization by AMF and Plant Total Fresh Weight

Intraradical AMF structures were assessed in *A. officinalis* plants treated with four different AMF species (*R. irregularis*, *R. intraradices*, *R. aggregatus* and *R. clarus*) both at the plants’ transfer in the containers (T0) and at the end of the experiment (i.e., after 9 days of growth in the system—T1). The overall data for AMF colonization and the total fresh weight (TFW, e.g., the sum of roots and shoots) at the two sampling times are reported in Appendix A. To correctly analyze the data, a mixed model for repeated measurements was conducted. For the AMF total colonization (TC%), no significant interaction (*p*-value = 0.0853) was found between “treatments” and “time”, while a significant effect (*p*-value < 0.001) was noticed for both single factors. The pairwise multiple comparison test revealed a general significant decrease in TC% (i.e., averaged of the four AMF treatments) between T0 and T1 (data not shown). A significantly lower TC% (i.e., average of the two sampling times for each AMF treatment) was also observed in plants associated with *R. irregularis* as compared to *R. intraradices*, while the plants associated with *R. clarus* and *R. aggregatus* had intermediate values (Table 1). Similarly, for the percentage of arbuscules (AC%), no significant interaction (*p*-value = 0.7479) was found between “treatments” and “time”, while a significant effect (*p*-value < 0.001) was noticed only for the single factor “time”. The pairwise multiple comparison test revealed a general decrease in AC% (i.e., average of the four AMF treatments) between T0 and T1 (data not shown).

The total fresh weight was measured on the same plants associated with the four AMF species at T0 and T1. No significant effect (*p*-value = 0.5699) was reported between “treatments” and “time”, or for the single factors (*p*-value = 0.3570 and *p* = 0.0897, respectively) (Table 1).

### 2.2. Metabolic Profiles and Metabolomic Analysis of A. officinalis Plants

Metabolic profiles of root and shoot samples of plants associated with four different AMF species were analyzed using multivariate analyses. The results of the unsupervised principal component analyses (PCA) highlighted the presence of two major and distinguished clusters (*R. irregularis*/*R. intraradices* on one side and *R. clarus*/*R. aggregatus* on the other side) in both the root and shoot samples (Figure 1a,b).

Shoot samples showed a higher proximity of individuals in cluster 2 (*R. clarus*/*R. aggregatus*) as compared to the corresponding root samples, which were more widespread in the model. Both root and shoot parts of the plants showed an outlier in cluster 1, in *R. irregularis* and *R. intraradices* treatments, respectively, and were subsequently removed from the analysis.

The UHPLC-HRMS metabolic analysis of root and shoot samples associated with the four AMF species gave, in total, 201 and 265 different mass signals, respectively, during the peak picking process. Based on the Volcano-plot analysis performed, only 96 and 101 compounds passed the defined criteria (fold change > 1.5 and *p*-value < 0.05) in all the possible AMF combinations tested (Figure 2 and Figure 3).

No differentiation was observed in treated root and shoot samples belonging to the same PCA cluster. However, significant up and down-regulation of mass signals were highlighted when comparing *R. irregularis* and *R. intraradices* treatments, from cluster 1 to *R. clarus* and *R. aggregatus* treatments from cluster 2 (Figure 2 and Figure 3). Among root and shoot samples, 22 and 35 mass signals, respectively, showed significant differences in all possible AMF-plants combinations between the two generated clusters. Thus, special attention was given and a detailed dereplication process was followed.

### 2.3. Identification of Primary and Secondary Metabolites Affected by AMF

The dereplication process of root and shoot extracts obtained from *A. officinalis* plants associated with the four different AMF species highlighted 42 differently modulated primary and secondary metabolites, strictly related to the specific AMF association.

As shown in Table 2a, six compounds were characterized as primary metabolites. Among them, compounds **1**, **2**, **3**, **6** were tentatively identified as amino acids, while compounds **4** and **5** as organic acids. Compounds **3** and **4** (glutamic and threonic acids, respectively) were overall increased in the shoots of the AMF-treated plants. Indeed, in the *R. intraradices* treatment, a significant induction of compound **3** was noticed and, together with *R. irregularis* treatment, of compound **4** in the shoots. Similarly, compound **1** (glutamine) was up-regulated in the roots associated with *R. irregularis* and *R. intraradices*.

Among the common compounds, which were affected in both parts of the plant and felt within our selection criteria (fold change > 1.5 and *p*-value < 0.05), compounds **5** and **6** showed the highest increment in plants associated with *R. irregularis*. In particular, compound **6** showed a maximum fold change of 8.66× and 7.37× in roots and shoots, respectively, while regarding compound **5**, the increment was slightly higher in the shoots (7.67×) as compared to the roots (6.85×). Conversely, compound **2** showed a 6.22× fold change in roots and a 5.87× fold change in shoots of *A. officinalis* associated with *R. intraradices*. Regarding the increment of compounds **1**, **3** and **4**, only those related to the roots, for compound **1** (1.7× fold change), and to the shoots, for compounds **3** and **4** (1.58× and 5.85× fold change, respectively) of the plants associated with *R. intraradices*, felt within our selection criteria (fold change > 1.5 and *p*-value < 0.05) (Figure 4).

The colonization of *A. officinalis* plants by the four different AMF species also impacted the secondary metabolites production in both parts of the plants (Figure 4 and Figure 5) (Table 2b). Overall, thirty-six secondary metabolites were characterized, belonging to different chemical classes of natural products. The majority of the identified compounds were assigned to phenylpropanoid derivatives (20) and to glycosidic triterpenes (9, saponins), while a less pronounced impact was observed in other chemical classes such as benzoic acids (1), secoiridoids (1), coumarins (2) and imidazolidines (1) (compounds **7**, **14**, **18**, **36**, **41**, **42**).

The phenolic C_6_-C_3_ derivatives represented the chemical group with the most important variation. In total, twenty phenylpropanoid derivatives were annotated. Fifteen of them were significantly increased in roots, nine in shoots, while three were affected in both roots and shoots (compounds **29**, **31** and **33**) (Figure 4 and Figure 5). Compound **29** was equally affected in both parts, while compounds **31** and **33** showed a stronger accumulation in shoots and roots, respectively. In all cases, a higher impact was observed in the *R. irregularis* and *R. intraradices* treatments (Figure 4 and Figure 5).

Among the C_6_-C_3_ derivatives, sixteen compounds were identified as mono-, di-, tri and tetrameric derivatives of caffeic acid (compounds **8**, **9**, **12**, **15**–**17**, **21**, **23**, **26**, **28**–**30**, **32**–**35**) and four of them were found in the corresponding glycosidic form (compounds **9**, **12**, **15**, **21**). Besides the hydroxycinnamates, compounds **13**, **24** and **31** were identified as derivatives of syringin and showed characteristic fragment ions at *m*/*z* 191, resulting from the cleavage of the glucose moiety, and fragment ions at *m*/*z* 176, 161, and 121 from the fragmentation of the remaining sinapyl alcohol [14]. In particular, compounds **13** and **24** were identified as the methyl derivatives of syringin and of syringinoside, already described in *A. officinalis* [14]. This was evident by the mass difference of 14 Da in their [M-H]^−^ pseudomolecular ions and by their MS/MS fragmentation ions at *m*/*z* 208 and 219, suggesting the presence of an additional methyl group (-CH_3_). Compound **31** showed a mass difference of 42 Da when compared to compound **24**. Both compounds shared similar MS/MS fragment ions (208, 191, 176, 121) suggesting their structural similarities to syringin. An additional diagnostic fragment ion at *m*/*z* 384 suggested the presence of an additional acetyl group (CH_3_CO) in the structure, which was assigned to the position C-6 of glucose moiety (Table 3). This leads to the identification of compound **31** as the acetyl derivative of compound **24** which represents a previously undescribed molecule in the literature. Compound **10** showed a mass difference of 30 Da with respect to syringin (sinapyl alcohol glucoside). The presence of an additional fragment at *m*/*z* 208, characteristic to a methyl ester group in the structure, and of the ion at *m*/*z* 193 allowed us to hypothesize the presence of a dihydrosinapic acid glucoside moiety. Further investigation of the MS/MS data and by comparison with bibliographic references [26], compound **10** was identified as a glycosylated methyl ester derivative of dihydrosinapic acid.

Regarding the glycosidic saponins, the four AMF species associated with *A. officinalis* modulated the expression of four di-glycosides (compounds **19**, **22**, **25** and **27**) and five tri-glycosides (compounds **20**, **37**–**40**) of oleanolic acid, mainly in the shoot of the plants (Figure 4). Indeed, six compounds were exclusively increased in the shoots (compounds **25**, **27**, **37**, **38**, **39**, **40**), while one was increased in both parts of the plants (compound **20**). This accumulation was noticed in plants associated with *R. irregularis* and *R. intraradices*.

Saponins **19** and **22** showed a molecular ion at *m*/*z* 828.4502 and at *m*/*z* 870.4608, respectively. Compound **19** was already reported as the oleanolic acid diglycoside anchusoside-9 presenting the two characteristic MS/MS fragments at *m*/*z* 665 and 503 resulting from the consecutive neutral loss of two glucose units (−162 Da and −324 Da). Further analysis of the MS/MS fragmentation pattern of compounds **19** and **22** showed common ions at *m*/*z* 503, 161, 113, 85, 71 corresponding to the aglycone hydroxybayogenin. Compound **22** presented the additional MS/MS fragment at *m*/*z* 707 resulting from the neutral loss of a hexose unit (−162 Da) and by the presence of an acetylated hexose unit esterified at the C-21 of hydroxybayogenin aglycone. These data lead to the tentative identification of compound **22** as the acetyl derivative of anchusoside-9 (C_44_H_69_O_17_) (Table 2b). Compound **22** represents a previously undescribed molecule in the literature.

Compound **25** showed a pseudomolecular ion *m*/*z* 839.4435 [M-H]^−^ and was tentatively annotated as a diglycoside derivative of bayogenin [33], while compound **20** showed a molecular ion at *m*/*z* 1001.4954 corresponding to the presence of an additional glycosidic unit in the structure (+162 Da) (Table 2b). The analysis of the MS/MS fragmentation pattern of both compounds showed a common fragment at *m*/*z* 633 corresponding to the cleavage of a glucuronic acid methyl ester, for compound **25**, and of a disaccharide moiety, containing a glucuronic acid methyl ester moiety and of an additional hexose unit, regarding compound **20**. Further, MS/MS fragments at *m*/*z* 797 and at *m*/*z* 633 for compounds **20** and **25**, respectively, derived from the neutral loss of the esterified sugar moiety at C-17, confirmed this hypothesis.

The MS/MS spectra of compound **27** (*m*/*z* 843.4406 [M-H]^−^) showed two major fragment ions at *m*/*z* 621 and 459 corresponding to the consecutive loss of a carboxyl unit at C-17 and of a glycosidic unit (−222 Da) following a further cleavage of the second hexose (−162 Da). The above-mentioned fragments suggested the presence of two additional hydroxyl groups in the aglycone with respect to bayogenin and leads to the tentative identification of compound **27** as a diglycosidic derivative of dihydroxybayogenin [35].

An increased production of compounds **37** to **40** was also noticed in shoots of *A. officinalis* associated with *R. irregularis* and *R. intraradicens*. Both compounds, **37** and **40**, showed a common molecular ion at *m*/*z* 1028.5187 and a common MS/MS fragmentation pattern, suggesting their structural similarity. Indeed, they both shared fragments at 779, 659, 617 and 455 (Table 3), characteristic of a tri-glycosylated configuration in both structures. In more detail, fragment ion at *m*/*z* 779 was obtained from the neutral loss of a hexoside unit and of a malonyl at C-6′ position, while the diagnostic fragment MS/MS ions at *m*/*z* 617 corresponded to the cleavage of an additional hexose unit. Further loss of the third sugar moiety at position C-3 offered the MS/MS ion at *m*/*z* 455 corresponding to the aglycone oleanolic acid. The position of the malonyl unit was established based on the diagnostic MS/MS fragment at *m*/*z* 659, corresponding to the loss of two hexoses (−324 Da) and of a carboxyl unit at C-17 (−46 Da) (Table 3). Based on the above-mentioned HRMS/MS data and by comparison with previously reported data [14,38], compounds **37** and **40** were tentatively assigned as isomers of anchusoside-2 and anchusoside-7. Compounds **37** and **40** presented a retention time of 8.15 and 9.26 min. Based on their calculated ClogP values of 5.15 and 4.67 for compounds **37** and **40**, respectively, the peak at 8.15 min was tentatively identified as the malonyl derivative of anchusoside-2 (compound **37**) [39], while the peak at 9.26 min as the malonyl derivative of anchusoside-7 (compound **40**). Both malonyl saponins represent previously undescribed molecules in the literature.

Compound **38** presented a molecular ion at *m*/*z* 1043.5081 and a chemical formula of C_51_H_80_O_22_. The HRMS/MS fragmentation pattern showed prominent characteristic ions at *m*/*z* 795, 659, 617 and 471, suggesting the similarity with compounds **37** and **40**. The mass difference of 17 Da with respect to compounds **37** and **40** suggested the presence of a hydroxylated oleanolic type aglycone in the structure, which was confirmed by the presence of the fragment ion *m*/*z* at 471 as well as of the fragment at *m*/*z* 795 generated by cleavage of one malonyl unit and of one hexoside. Accordingly, compound **38** was assigned as the hydroxyl malonyl derivative of anchusoside-7. Compound **39** showed a pseudomolecular ion at *m*/*z* 1129.5087 [M-H]^−^ and the diagnostic MS/MS fragments at *m*/*z* 659, 471 and 455. The mass difference of 86 Da with respect to compound **38** was attributed to the presence of an additional malonyl moiety leading to its tentative identification as the hydroxy di-malonyl derivative of anchusoside-2/7. Both compounds **38** and **39** represent previously undescribed molecules in the literature.

## 3. Discussion

The association between plants and arbuscular mycorrhizal fungi (AMF) is one of the most widespread symbioses [5]. These fungi provide numerous benefits to the host plants, especially in terms of nutritional assistance and resistance to a/biotic stresses [5,40]. A growing body of studies has reported the beneficial effects of these root symbionts on the modulation of specific biosynthetic pathways increasing/modifying the production of primary and secondary metabolites [2,8,9,14].

Herein, an untargeted metabolomic study on several AMF species belonging to the same genus was conducted under the highly-controlled semi-hydroponic cultivation system developed by Cartabia et al. [14], to evaluate the effects of four different AMF species (*R. irregularis*, *R. intraradices*, *R. clarus* and *R. aggregatus*) on the metabolome of *A. officinalis.*

During the experimental period, the root colonization was high for the four AMF species, even if a general significant decrease was noticed after 9 days of growth in the semi-hydroponic cultivation system (i.e., total colonization mean values varying between 85% at T0 to 66% at T1, and the percentage of arbuscules above 10% and close to 10% at T0 and T1, respectively).

### 3.1. Impact of AMF Species on Primary and Secondary Metabolites in Roots and Shoots of Anchusa officinalis

The major impact of AMF on *A. officinalis* metabolome was detected in the primary metabolism, mainly in the amino acid and organic acid content (compounds **1**–**6**), but also in some specific secondary metabolites, derived from the phenylpropanoid (compounds **8**–**10**, **12**, **13**, **15**–**17**, **19**, **21**, **23**, **24**, **26**, **28**, **29**–**35**) and the mevalonate (compounds **19**, **20**, **22**, **25**, **27**, **37**–**40**) pathways. The untargeted metabolomic approach performed on root and shoot tissues of *A. officinalis* evidenced forty-two compounds that fulfilled the defined threshold (fold change > 1.5 and *p*-value < 0.05) applied in the Volcano-plot analysis in at least one of the associations between the AMF species and *A. officinalis*.

#### 3.1.1. Impact on Primary Metabolism

Six primary metabolites (compounds **1**–**6**) were significantly affected in root and shoot samples in relation to the four AMF species. While compound **1** (glutamine) was produced in significantly higher amounts in the roots, compounds **2**–**6** were mainly accumulated in the shoots. All compounds (**1**–**6**) showed a similar accumulation increment in plants associated with *R. irregularis* and *R. intraradices* (cluster 1). *R. irregularis* and, especially, *R. intraradices* are strong elicitors of amino acids, such as aspartic (compound **2**), glutamic acid (compound **3**), glutamine (compound **1**) and its derivative, pyroglutamic acid (compound **6**), as well as of organic acids, such as threonic acid (compound **4**) and malic acid (compound **5**), as compared to *R. clarus* and *R. aggregatus*. Based on these results, *R. clarus* and *R. aggregatus*, belonging to cluster 2 (Figure 1), affect the primary metabolism less than the two other AMF strains belonging to cluster 1 (Figure 1).

Among the primary metabolites, compounds **2** and **3** have been reported as N precursors and donors, reflecting the AMF’s ability to enhance ammonium assimilation in mycorrhized plants by the GS/GOGAT enzymatic pathway [41]. Together with compound **5**, they are also important intermediate of the tricarboxylic acid cycle and they act as precursors for the synthesis of key amino acids, such as asparagine, threonine, lysine, isoleucine and glutamine (compound **1**), which are building blocks for the production of macromolecules [14,41]. Pyroglutamic acid (compound **6**), reported as the lactam of glutamic acid, is considered an important reservoir of glutamate [42] while compound **4** (threonic acid) is linked to ascorbic acid metabolism and catabolism, involved in anti-oxidant activities and correlated to the well-maintaining of plant fitness [43]. Threonic acid was already mentioned in previous studies as a naturally occurring constituent of shoots [44,45], and its modulation in AMF-plant symbiosis was mentioned by Schweiger et al. [11]. These results are in accordance with Cartabia et al. [14] who showed an important accumulation of the above-mentioned compounds in *A. officinalis* plants associated with *R. irregularis*.

#### 3.1.2. Impact on Secondary Metabolism

Thirty-six secondary metabolites emerged from the Volcano-plot analysis of root and shoot samples, as the most affected compounds by the colonization of *A. officinalis* with the four different AMF species. The annotated compounds were divided into four major categories: (1) twenty C_6_-C_3_ derivatives, from which four were classified as syringin derivatives and sixteen as caffeic acid derivatives; (2) one C_6_-C_2_ derivative; (3) nine glycosylated triterpenoids; (4) six compounds belonging to other chemical classes, such as secoiridoids, coumarins, and imidazolidines. Among them, eight compounds were affected both in roots and shoots of the AMF-colonized plants in one, at least, symbiotic association (compounds **14**, **20**, **29**, **31**, **33**, **34**, **41**, **42**).

Phenylpropanoids were the most impacted secondary metabolites class by the AMF treatments. They were characterized as mono, di, tri or tetrameric derivatives of caffeic acid and of syringin according to their characteristic MS and MS/MS fragments (see Section 2.3). Monomers and dimers of caffeic acid derivatives share characteristic fragment ions at *m*/*z* 179, 161, 135, 121, deriving from the cleavage of a single C_6_-C_3_ unit, while the tri- and tetrameric forms show additional ions at *m*/*z* 339, 295 and 185 derived from the cleavage of multiple units. Our results are in line with reported data suggesting a discernible enhancement of phenylpropanoid pathway in the roots of mycorrhized plants [2,10,14]. In addition, these compounds were already reported in *A. officinalis* [14,46].

All the identified C_6_-C_3_ and C_6_-C_2_ compounds were found in significantly higher amounts in plants associated with *R. irregularis* and *R. intraradices* (cluster 1) as compared to those associated with *R. clarus* and *R. aggregatus* (cluster 2). This suggests that *R. irregularis* and *R. intraradices* could influence *A. officinalis* metabolome in a similar way by activating common biosynthetic pathways. However, in a few cases, such as compounds 11 and 16 in shoots, their amount was exclusively affected (*p*-value < 0.05) in plants associated with *R. intraradices*. Besides the close response on metabolite productions by *R. irregularis* and *R. intraradices* species, minor differences can be observed in triggering specific compounds. In fact, AMF species belonging to the same cluster in the principal component analysis (PCA) affected *A. officinalis* plants in a similar way, however, not strictly identical. On the other hand, AMF-plant associations, which are differently clustered and present major differences in affecting *A. officinalis* metabolome, could equally affect the accumulation of specific secondary metabolites. This is the case of compounds **12**, **23**, **31** and **32** in roots, and compounds **11**, **15** and **16** in shoots, which did not show a significant accumulation (*p* < 0.05) among the AMF species belonging to different clusters.

The AMF species also affect the production of oleanane-type saponins. In our analysis, and in contrast to the phenylpropanoids derivatives, these compounds were essentially affected/modulated in the shoots of *A. officinalis*. Saponins are involved in plant defense mechanisms against biotic constraints, such as pest or herbivores attack, and their content is strongly influenced by plant-AMF symbiosis [47,48,49]. Nine significantly modulated compounds from the different AMF treatments were tentatively identified. Our analysis led to the identification of four oleanane-type derivatives, which possessed two glycosidic units in their configuration (compounds **19**, **22**, **25** and **27**) and five tri-glycosylated derivatives of bayogenin (compounds **20**) and of oleanolic acid (compounds **37**–**40**). The tri-glycosylated compound **20** and **25** presented a similar glycosylation, with the presence of a glucuronic methyl ester group, observed for the first time in the study of Cartabia et al. [14]. The ability of *A. officinalis* to produce saponins was already reported in previous studies [31,38,50] and their strong accumulation in shoot parts of mycorrhized *A. officinalis* is in line with the study by Cartabia et al. [14].

Seven compounds (**20**, **25**, **27**, **37**–**40**) were essentially modulated in the shoots of plants associated with *R. irregularis* and *R. intraradices* (cluster 1) as compared to the plants associated with *R. clarus* and *R. aggregatus* (cluster 2), from which six are exclusively identified in the shoots. On the other hand, Volcano-plot analysis of root parts showed the accumulation of two saponins (compounds **19** and **22**). Similarly, to the phenylpropanoid derivatives, triterpenoids are mainly affected during the association of *A. officinalis* with *R. irregularis* and *R. intraradices* (cluster 1). However, within cluster 1, compounds **37** and **40** appeared to be more affected in the shoots of plants associated with *R. intraradices* (*fold change* of 1.68 and 1.60, respectively). When compared to the AMF of cluster 2, *R. irregularis* failed to exert any significant upregulation of compound **25** in the shoots, while *R. intraradices* failed to induce any significant effect of compound 19 in the roots. In all the other cases, AMF belonging to cluster 1 significantly affected triterpenoids accumulation.

Special attention was given to compounds **37**–**40**, which represent the most affected saponins of plants associated with *R. irregularis* and *R. intraradices* (*fold change* ranking from 1.34 to 4.55). They are all undescribed molecules in the literature, characterized by a conjugated malonyl-sugar moiety and they represent derivatives of anchusoside-7 and anchusoside-2 [14,31,38]. Our results are in line with similar conjugated structures identified from the association of different plants with *R. irregularis* [2,9]. Kobayashi et al. [51] reported the presence of FAS II gene, responsible for the synthesis of lipoic acid through the mitochondrial pathway in bacteria, in both *R. irregularis* and *R. clarus* from which some subunits are encoding for enzymes such as malonyl-CoA ACP transacylase. This could explain the potential ability of AMF to upregulate and synthesize malonyl conjugated compounds.

The four AMF species also promoted, differently, the accumulation of acetylated compounds. This is the case of compound **19**, characterized as the acetyl derivative of anchusoside-9 (compound **22**), already reported in *A. officinalis* [14]. The mass difference between those two compounds, equal to the presence of an additional acetyl group, as well as to the presence of the diagnostic MS/MS fragment ions at *m*/*z* 707, deduced an acetylated hexose unit at the C-21 of the structure [28]. Compound **19** represents a previously undescribed molecule in the literature. Besides saponins, two additional compounds from the phenylpropanoid pathway, both induced in plants associated with *R. irregularis* and *R. intraradices* (compounds **12** and **31**) presented a similar acetyl-sugar conjugation (see Section 2.3). Compound **31** was tentatively identified as a new acetylated derivative of methylsyringin, while compound **12** was the 3-feruloyl-6′-acetyl sucrose [28]. To the best of our knowledge, these results pointed out, for the first time, the ability of specific AMF species to enhance the production of acetylated secondary metabolites.

The accumulation of methylated compounds was also reported in our analysis, essentially in the root parts of *A. officinalis* associated with *R. irregularis* and *R. intraradices*. Indeed, a significant increment of methylated syringin derivatives (compounds **13**, **24**, **31**) and of the methylated phenylpropanoids, methyl dihydrosinapic acid glucoside (compound **10**) and methylrosmarinic acid (compound **35**), was observed. This result is consistent with our previous study [14] pointing out the methylation potential of AMF *R. irregularis*.

Despite the widely accepted fact that different AMF genera could affect plant metabolome differently [52], one of the main observations of the present study is that AMF species belonging to the same genus may induce similar, but not strictly identical, metabolomic responses in *A. officinalis* plants, without being strongly related phylogenetically. Indeed, the latest updates regarding the phylogenetic classification of under-investigated AMF strains showed that *R. irregularis* is phylogenetically more closely related to *R. clarus* than to *R. intraradices* [53]. Therefore, the outcome of the association in terms of, e.g., plant growth promotion and metabolites enhancement, is highly specific to the identity of the AMF symbiont [2,54,55].

## 4. Material and Methods

### 4.1. Chemicals

Methanol (MeOH) HPLC grade was purchased from Fisher Chemical (Fisher Scientific, Loughborough, UK), ethyl acetate (EtOAc) (ExpertQ^®^, 99.8%) from Scharlau Basic (a.r. grade, Scharlab S.L., Barcelona, Spain) while acetonitrile (ACN) LC-MS grade (LiChrosolv^®^ hypergrade) and formic acid (FA) LC-MS grade (LiChropur^®^) were purchased from Merck (Merck KGaA, Darmstadt, Germany). The ultrapure water was obtained from a LaboStar apparatus (Evoqua LaboStar^®^ 4, Evoqua Water Technologies, Pittsburgh, PA, USA).

### 4.2. Biological Material

Seeds of *Anchusa officinalis* L. were provided by Rühlemann’s herbs and aromatic plants (Germany). They were surface-disinfected by immersion in sodium hypochlorite (8% active chloride) for 5 min and rinsed three times with sterilized (121 °C for 15 min) deionized water. The seeds were then germinated in plastic seed trays (37.5 × 23 × 6 cm) filled with a mix (*w/w*, 1:2) of sterilized (121 °C for 15 min) perlite (Perligran Medium, KNAUF-GMBH, Dortmund, Germany) and turf (DCM, Grobbendonk, Belgium). The trays were placed in the greenhouse set at 25 °C/18 °C (day/night), a relative humidity (RH) of 38%, a photoperiod of 16 h day^−1^ and a photosynthetic photon flux (PPF) of 120 μmol m^−2^ s^−1^.

Four AMF species were supplied by the *Glomeromycota* in vitro collection (GINCO) (http://www.mycorrhiza.be/ginco-bel/, accessed on 17 June 2022); *Rhizophagus irregularis* (Błaszk, Wubet, Renker and Buscot) C. Walker and A. Schüßler as [‘irregulare’]) MUCL 41833, *Rhizophagus intraradices* (N.C. Schenck & G.S. Sm.) C. Walker & Schuessler) MUCL 49410, *Rhizophagus clarus* (T.H. Nicolson & N.C. Schenck) C. Walker & A. Schüßler) MUCL 46238 and *Rhizophagus aggregatus* (N.C. Schenck & G.S. Sm.) C. Walker MUCL 49408. The fungi were proliferated on plants of *Zea mays* L. cv. ES Ballade (Euralis, Lescar, France) in a 10-L plastic box containing sterilized (121 °C for 15 min) lava (DCM, Grobbendonk, Belgium). The plants were grown under the same greenhouse conditions as above.

### 4.3. Anchusa officinalis Colonization

Two-week-old *A. officinalis* plants were transferred in 10 L pots containing a sterilized (121 °C for 15 min) mix of lava and perlite (*w*/*w*, 2:1). The substrate was half mixed with the AMF-inoculum substrate above (final-ratio lava:perlite *w*/*w*, 5:1). The plants were grown under the same greenhouse conditions as above.

### 4.4. Experimental Setup

Two-month-old plants (7 replicates per AMF treatment) were gently removed from the 10 L pots above and their root systems were rinsed with deionized water to eliminate lava and perlite debris. They were subsequently transferred to the semi-hydroponic cultivation system as detailed in Cartabia et al. [14] (Figure 6). Briefly, the plants were placed in a 500 mL plastic bottle (VWR INTERNATIONAL, Leuven, Belgium), cut at the base and with a 100 μm size pore nylon mesh (Prosep B.V.B.A., Zaventem, Belgium) glued on the top. The bottles (called containers thereafter) were used bottom-up, filled with 32 g of perlite (KNAUF GMBH, Dortmund, Germany), covered with a superficial layer of black lava rock (1–3 mm) and wrapped in aluminum foil to avoid algae development. The containers were transferred randomly in holes made in flex-foam supports and were maintained in the greenhouse set at the same conditions as described above. A 90% P-impoverished modified Hoagland solution (see [56]) was used at two different concentrations: diluted by 200× (referred to as Hoagland^dil200×^) during the acclimatization phase (7 days) and diluted by 100× (referred to as Hoagland^dil100×^) during the circulation phase (42 h). Both phases were presented in detail in Cartabia et al. [14]. Before starting with the circulation, initial flushing was performed. Each plant container received 200 mL of Hoagland^dil100×^ solution, which was circulated at a velocity of 44 mL min^−1^ through the containers and then discarded. After this initial flushing, regular circulation was initiated and maintained at 7.5 mL min^−1^ for 42 h (T1).

### 4.5. Plant Harvest and AMF Roots Colonization

Total fresh weight (TFW), as well as root colonization, were assessed on the same 7 replicates per AMF treatment at the start (T0) and the end of the experiment (T1). The root colonization was evaluated by McGonigle et al. [57], on one-third of the root system (i.e., 1/6 of the root system at T0 and 1/6 at T1). The root fragments were first stained, following the method developed by Walker [58], and subsequently placed on microscope slides and covered with a 40 × 22 mm coverslip before observation under a bright field light microscope (Olympus BH2-RFCA, Japan) at ×10 magnification. Around 200 intersections were observed per plant to evaluate the total colonization (TC%) of roots (e.g., hyphae, arbuscules, and vesicles) and the percentage of arbuscules (AC%) (Figure 6). At the end of the circulation above (T1), the plants were harvested to proceed with the primary and secondary metabolites analysis on the other two-thirds of the root system (see below).

### 4.6. Analysis of Primary and Secondary Metabolites in Roots and Shoots of A. officinalis

#### 4.6.1. Samples Preparation

The remaining two-thirds of the root systems, as well as the shoot parts of each plant, were prepared according to Cartabia et al. [14]. Briefly, 20 mg of powdered and freeze-dried plant material was subjected to an exhaustive ultrasound-assisted extraction with 1 mL of EtOAc/MeOH (35:65, *v*/*v*) mixture for 30 min at 25 °C. The samples were centrifuged at 3.500 rpm for 3 min. The supernatants obtained from 3 cycles of extraction were combined and dried under a gentle nitrogen stream. Dried extracts were re-dissolved in H_2_O: MeOH (50:50, *v*/*v*) of LC-MS grade to obtain a final concentration of 300 μg mL^−1^ and filtered through a 45 μm pore size hydrophilic polyvinylidene fluoride membrane prior to UHPLC-HRMS analysis. Each plant was analyzed in triplicate.

#### 4.6.2. UHPLC-HRMS Analysis and Untargeted Metabolomics Data Processing

UHPLC-HRMS analysis and MS/MS data processing of root and shoot samples of the plants associated with *R. irregularis* MUCL 41833, *R. intraradices* MUCL 49410, *R. clarus* MUCL 46238 and *R. aggregatus* MUCL 49408 were processed according to Cartabia et al. [14], with some minor modifications. The data acquisition was performed on the HRMS/MS Orbitrap Q-Exactive platform (Thermo Scientific, San Jose, CA, USA) in the full scan ion mode with a mass range of 100–1200 Da. The HRMS data were collected in negative ionization mode applying a resolution of 70.000 on a centroid mode. The conditions for the HRMS negative ionization mode were the following: capillary temperature, 320 °C; spray voltage, 2.7 kV; S-lense Rf level, 50 V; sheath gas flow, 40 arb. units; aux gas flow, 5 arb. units; aux. gas heater temperature, 50 °C. The HRMS/MS spectra were recorded for the three most intense ion peaks with a threshold of a 10 s dynamic exclusion at a resolution of 35.000. The stepped normalized collision energy was set at 40, 60, and 100. A Hypersil Gold UPLC C18 (2.1 × 100 mm, 1.9 μm) reverse phase column (Thermo Fisher Scientific, San Jose, CA, USA) was used for the separations and the mobile phase consisted of solvents: A ultra-pure H_2_O 0.1% (*v*/*v*) FA and B ACN. A 16 min gradient method for the elution of compounds was set up as follows: T = 0 min, 5% B; T = 1, 5% B; T = 11 min, 95% B; T = 14 min, 95% B (column cleaning); T: 14.1 min, 4% B; T = 14.6 min, 5% B; T = 16 min, 5% B (column equilibration). The flow rate applied for the analyses was 0.260 mL min^−1^ and the injection volume μ. The column temperature was kept at 40 °C while the sample tray temperature was set at 10 °C.

In the following steps, an untargeted metabolomics workflow, including the normalization of the dataset (deconvolution, deisotoping, retention time -RT- alignment and gap-filling procedures), was developed for the detection of known and unknown compounds. All the raw data obtained from the high-resolution metabolomic profiling were uploaded in Compound Discoverer 3.2.0.421 software (Thermo Fisher Scientific, San Jose, CA, USA). Briefly, the peak alignment of the selected spectra was performed from 1 to 12 min with a mass tolerance (MT) of 5 ppm and a maximum shift of 2 min. The spectrum properties applied for the peak picking and detection of compounds were the following: S/N > 3, Min. peak intensity of 7.5 × 10^5^ and MT < 5 ppm, and the integration of selected adducts ions for ESI^−^ ionization ([2M+FA-H]^−1^; [2M-H]^−1^, [M+FA-H]^−1^, [M-H]^−1^; [M-H_2_O]^−1^). Finally, the grouping of compounds was performed with an MT < 5 ppm and an RT tolerance of 0.5 min.

The structural elucidation of the metabolites of interest was performed by comparison of the chromatographic and spectrometric features of each respective peak with data from the literature. The high resolving power for both full scan experiments and the MS/MS fragments of the Q-Exactive Orbitrap analyzer in correlation to the accurate mass measurements assured the identification of the very important variables (VIP) compounds with high confidence. The suggested EC (Elemental Composition) for molecular ions and MS/MS fragments, as well as the respective RDBeq (Ring Double Bond Equivalents) further assisted the safe identification process. More prediction of compounds was performed by comparing data with in-house and online libraries, and fragment ions were correlated with spectra online databases. The pre-processed data of ESI^−^ ionization were exported as a .csv file to a Microsoft Excel spreadsheet and manipulated accordingly for the data filtering and the multivariate statistical analysis followed.

### 4.7. Statistical Analysis

AMF colonization parameters (TC% and AC%) were subjected to a mixed model for repeated measurements fit by restricted maximum likelihood estimation. The normal distribution of residuals was checked for each dependent variable. The model took into account the heterogeneity of the variance (only for TC%) modeling the within-group errors variance structure with the “varIdent” matrix and assuming the different AMF treatments as a stratification variable [59]. Moreover, the repeated measurements (i.e., two sampling times conducted on the same replicates) were modeled through an autoregressive correlation structure of order 1 (“corAR1”) for all the dependent variables (TC% and AC%). An ANOVA test of the model was provided and the interaction between “time” (T0 and T1) and “treatments” (*R. irregularis*, *R. intraradices*, *R. clarus* and *R. aggregatus*) was checked as well as the significance of every single factor separately. A pairwise multiple comparison test (with the Bonferroni correction) was computed to separate means (*p*-value < 0.05). Similarly, the total fresh weight of the plants was subjected (as a dependent variable) to the same mixed model as described above. Normal distribution of residuals and homogeneity of variance was checked. The model took into account the repeated measurements (i.e., two sampling times conducted on the same replicates) through an autoregressive correlation structure of order 1 (“corAR1”). Data analyses were performed by R [60] using the “nlme” package [61].

Multivariate analyses of HRMS data were carried out using SIMCA 14.1 software (Soft Independent Modelling of Class Analogy) (Umetric, Malmo, Sweden) to assign the discriminant metabolic changes between the different AMF treatments (*R. irregularis*, *R. intraradices*, *R. clarus* and *R. aggregatus*) after 9 days of the experiment. The interpretation of imported data was performed through a principal component analysis (PCA) and partial least squares discriminant analyses (PLS-DA) according to Pareto correlation. In addition, a permutation test with *n* = 100 was performed to exclude any overfitting of the aforementioned PLS-DA models (Appendix A). Volcano-plot analyses were carried out using Compound Discoverer 3.2.0.421 (Thermo Fisher Scientific, San Jose, CA, USA) on the basis of filtering criteria such as *p*-value < 0.05 and fold change > 1.5, while the graphical representation (bar charts) of discriminant variables (targeted compounds) were generated with GraphPad Prism 7 (GraphPad Software, San Diego, CA, USA). One-way ANOVA and Tukey’s test were provided in order to reveal significant differences (*p*-value < 0.05) in the discriminant metabolites between the four AMF treatments (*R. irregularis*, *R. intraradices*, *R. clarus* and *R. aggregatus*). Data analyses were performed by R (R Core Team, 2018) using “ggplots 2” [62] and “agricolae” [63] packages. Finally, the identification and matching of discriminant variables were performed by comparing the MS and MS/MS spectra with bibliographic data as well as with commercial and in-house libraries.

## 5. Conclusions

In the present study, we demonstrated that the association between *Anchusa officinalis* and different AMF species (*Rhizophagus irregularis*, *R. intraradices*, *R. clarus* and *R. aggregatus*) belonging to the same genus resulted in a different modulation of several metabolites. Based on our data, primary and secondary metabolites production was significantly affected especially in the plants associated with *R. irregularis* and *R. intraradices*. Indeed, a higher accumulation of phenolic compounds and of saponins was detected in roots and shoots of *A. officinalis* plants associated with these two AMF species. Additionally, an increased production of malonyl, acetyl and methyl derivatives of phenylpropanoids (e.g., 3-feruloyl-6′acetyl sucrose, methylsyringinoside, methylsyringin, 6″-acetylmethylsyringin) and of oleanane-type saponins (e.g., acetylanchusoside-9, malonylanchusoside-2, malonylanchusoside-7) was observed. Among them, six compounds (acetylanchusoside-9, 6″-acetylmethylsyringin, malonylanchusoside-2, hydroxy-malonylanchusoside-7, hydroxy-dimalonylanchusoside2/7 and malonylanchusoside-7) were tentatively characterized as new secondary metabolites. Within this study, evidence leads to the AMF species-specific metabolic response of *A. officinalis*. However, some AMF may be more closely related to each other in modulating the plant metabolome of their host. These observations may open the door to the selection of the most adequate AMF species and/or strains for the production of desirable active compounds.

## Figures and Tables

**Figure 1 metabolites-12-00573-f001:**
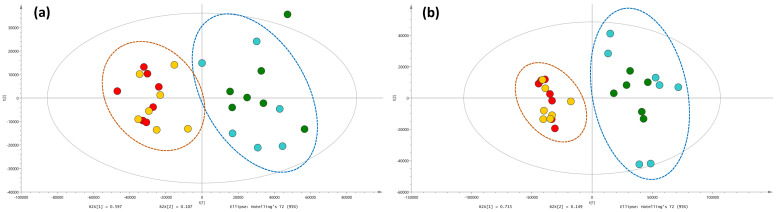
Principal component analysis (PCA)—Comparison of UHPLC-HRMS metabolic profiles from *A. officinalis* root (**a**) and shoot (**b**) samples associated with *R. irregularis*, *R. intraradices*, *R. clarus* and *R. aggregatus*, after 9 days of growth in the semi-hydroponic cultivation system. (*R. irregularis* MUCL 41833: blue dots; *R. intraradices* MUCL 49410: green dots; *R. clarus* MUCL 46238: red dots; *R. aggregatus* MUCL 49408: yellow dots).

**Figure 2 metabolites-12-00573-f002:**
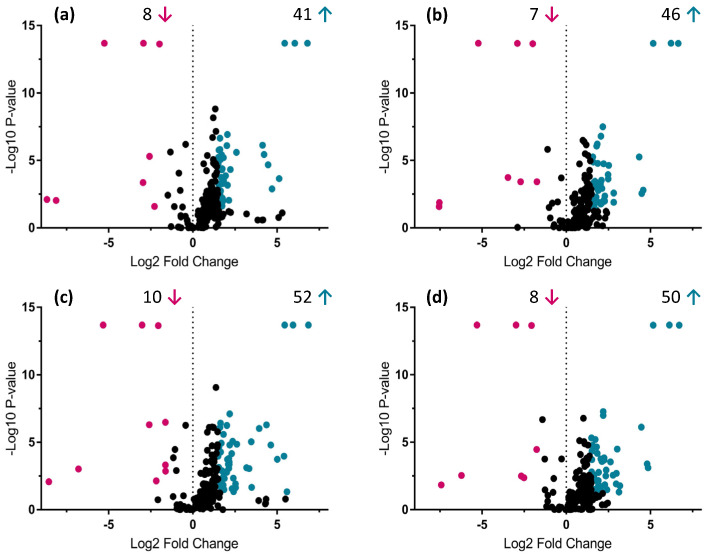
Volcano-plot analysis—Identification of up- and down-regulated compounds (*p*-value < 0.05 and fold change > 1.5) between *A. officinalis* root samples associated with four AMF species (*R. irregularis*, *R. intraradices*, *R. clarus* and *R. aggregatus*) after 9 days of growth in the semi-hydroponic cultivation system. Comparison of metabolic profiles from root samples associated with (**a**) *R. irregularis* MUCL 41833 and *R. clarus* MUCL 46238; (**b**) *R. irregularis* MUCL 41833 and *R. aggregatus* MUCL 49408; (**c**) *R. intraradices* MUCL 49410 and *R. clarus* MUCL 46238; (**d**) *R. intraradices* MUCL 49410 and *R. aggregatus* MUCL 49408. Significant up-regulated compounds are represented in blue (right side of the plots) and down-regulated in magenta (left side of the plots). Blue and magenta arrows represent the amount of up- and down-regulated compounds, respectively, in the specific AMF-plants treatment.

**Figure 3 metabolites-12-00573-f003:**
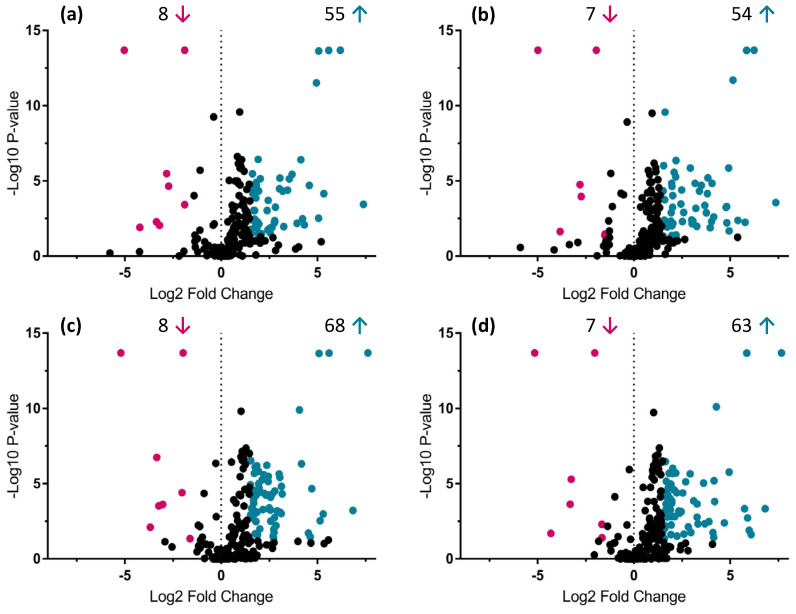
Volcano-plot analysis—Identification of up- and down-regulated compounds (*p*-value < 0.05 and fold change > 1.5) between *A. officinalis* shoot samples associated with four AMF species (*R. irregularis*, *R. intraradices*, *R. clarus* and *R. aggregatus*) after 9 days of growth in the semi-hydroponic cultivation system. Comparison of metabolic profiles from shoot samples associated with (**a**) *R. irregularis* MUCL 41833 and *R. clarus* MUCL 46238; (**b**) *R. irregularis* MUCL 41833 and *R. aggregatus* MUCL 49408; (**c**) *R. intraradices* MUCL 49410 and *R. clarus* MUCL 46238; (**d**) *R. intraradices* MUCL 49410 and *R. aggregatus* MUCL 49408. Significant up-regulated compounds are represented in blue (right side of the plots) and down-regulated in magenta (left side of the plots). Blue and magenta arrows represent the amount of up- and down-regulated compounds, respectively, in specific AMF-plants treatment.

**Figure 4 metabolites-12-00573-f004:**
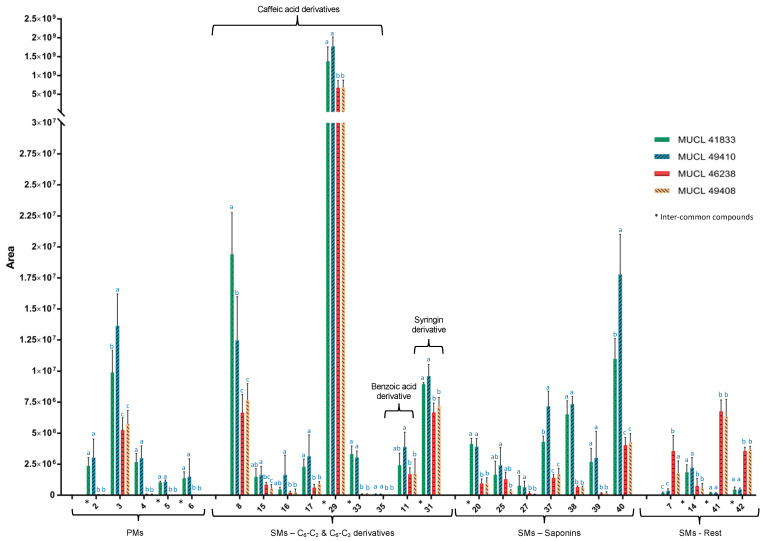
Graphical representation of metabolome profile variations in shoots of *Anchusa officinalis* associated with four different AMF species (*R. irregularis* MUCL 41883, *R. intraradices* MUCL 49410, *R. clarus* MUCL 46238 and *R. aggregatus* MUCL 49408). The AMF treatment means followed by the same lowercase letters are not significantly different according to HSD Tukey’s test (*p*-value < 0.05).

**Figure 5 metabolites-12-00573-f005:**
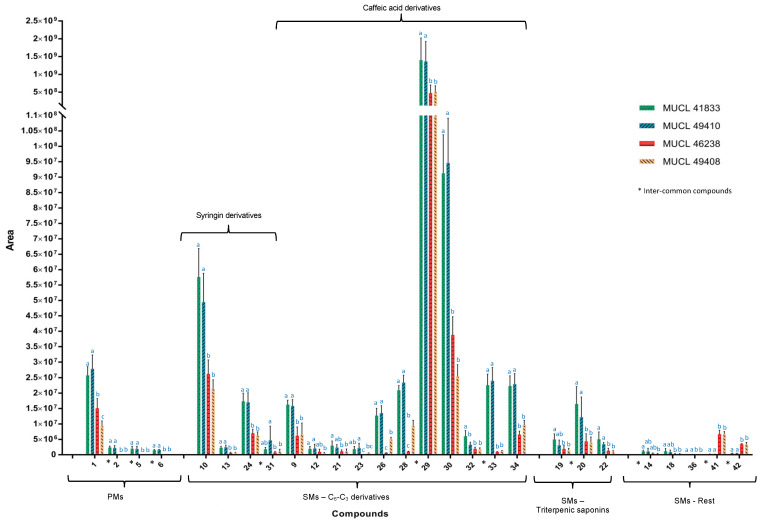
Graphical representation of metabolome profile variations in roots of *Anchusa officinalis* associated with four different AMF species (*R. irregularis* MUCL 41883, *R. intraradices* MUCL 49410, *R. clarus* MUCL 46238 and *R. aggregatus* MUCL 49408). The AMF treatment means followed by the same lowercase letters are not significantly different according to HSD Tukey’s test (*p*-value < 0.05).

**Figure 6 metabolites-12-00573-f006:**
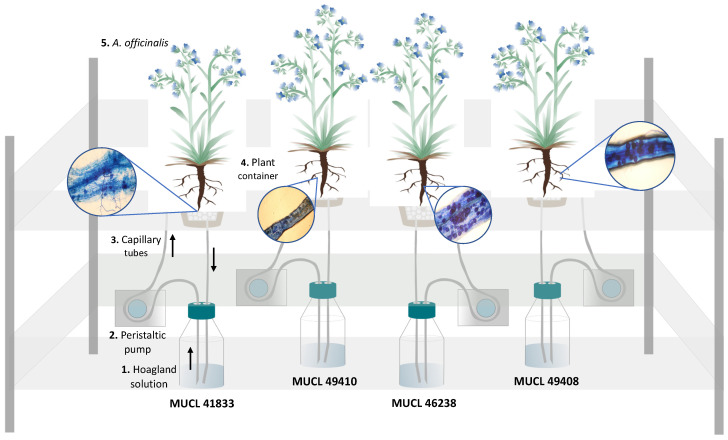
Schematic representation of the circulatory semi-hydroponic cultivation system. The Hoagland solution circulated through the containers supporting *Anchusa officinalis* plants associated with four different AMF species (*R. irregularis* MUCL 41833; *R. intraradices* MUCL 49410; *R. clarus* MUCL 46238; *R. aggregatus* MUCL 49408). The nutrient solution in the glass bottle (1) is pumped using a peristaltic pump (2) via silicon tubes (3) to the upper part of the plant container (4) containing *A. officinalis* plants (5). The solution percolates through the plant container back into the glass bottle. The black arrows indicate the flow direction of the nutrient solution in the tubing. The roots-stained images represent the plant-AMF colonization of the four different AMF species applied in this study.

**Table 1 metabolites-12-00573-t001:** AMF-root colonization (TC% and AC%) and total fresh weight (TFW) averaged between the two-time samplings (T0 and T1) of *A. officinalis* associated with each AMF species (*R. irregularis*, *R. intraradices*, *R. clarus* and *R. aggregatus*) growing for 9 days in the semi-hydroponic cultivation system.

AMF Treatments	AMF Root Colonization (%)		Fresh Weight (g)
	TC	AC	
*R. irregularis* (MUCL 41833)	70 ± 2 a	10 ± 3 a	5.68 ± 1.5 a
*R. intraradices* (MUCL 49410)	81 ± 2 b	17 ± 3 a	8.22 ± 1.5 a
*R. clarus* (MUCL 46238)	74 ± 2 ab	12 ± 3 a	5.13 ± 1.5 a
*R. aggregatus* (MUCL 49408)	77 ± 2 ab	14 ± 3 a	8.06 ± 1.5 a

The parameters measured are expressed as mean ± standard errors of 14 replicates per each AMF treatment. The AMF treatment means followed by the same lowercase letters are not significantly different according to Bonferroni post-hoc test (*p*-value < 0.05).

**Table 2 metabolites-12-00573-t002:** Affected primary (**a**) and secondary metabolites (**b**) in root and shoot samples of *Anchusa officinalis* associated to *R. irregularis* MUCL 41883, *R. intraradices* MUCL 49410, *R. clarus* MUCL 46238 and *R. aggregatus* MUCL 49408 growing for 9 days in the semi-hydroponic cultivation system.

	Peak	Proposed Phytochemicals	R_t_ (min)	Precursor Ion—[M-H]^−^	*m*/*z* Calcd.	Δm (ppm)	MS/MS Fragment Ions (*m*/*z*)	Chemical Formula	Affected In	Ref.
**(a)** PMs	1	D-Glutamine	1.43	145.0620	146.0686	1.13	146, 128, 102	C_5_H_10_N_2_O_3_	R	[20]
	2	L-Aspartic acid	1.44	132.0303	133.0370	0.67	132, 115, 88, 71	C_4_H_7_NO_4_	S, R	[20]
	3	L-Glutamic acid	1.46	146.0660	147.0530	0.75	146, 128, 102	C_5_H_9_NO_4_	S	[20]
	4	L-Threonic acid	1.51	135.0300	136.0366	1.06	135, 117, 89, 75, 61	C_4_H_8_O_5_	S	[20]
	5	DL-Malic acid	1.59	133.0144	134.0210	0.93	133, 115, 89, 72, 71	C_4_H_6_O_5_	S, R	[21]
	6	DL-pyroglutamic acid	1.65	128.0355	129.0420	1.28	128, 82, 62	C_5_H_7_NO_3_	S, R	[22]
**(b)** SMs	7	Allantoin	1.49	157.0359	158.0434	−1.56	114, 97, 71, 59	C_4_H_6_N_4_O_3_	S	[23]
	8	Danshensu	3.53	197.0451	198.0523	0.64	179, 153, 135, 121, 73	C_9_H_10_O_5_	S	[24]
	9	Glomeratose A	4.50	561.1837	562.1892	2.19	342, 240, 191, 163, 121, 59	C_24_H_34_O_15_	R	[25]
	10	Methyl dihydrosinapic acid glucoside	4.90	401.1458	402.1520	1.15	208, 193, 175, 163, 121, 93, 71	C_18_H_26_O_10_	R	[26]
	11	Salicylic acid glucoside	4.94	299.0776	300.0840	1.50	137, 93	C_13_H_16_O_8_	S	[27]
	12	3-Feruloyl-6′acetyl sucrose	5.05	559.1679	560.1736	2.16	193, 179, 161, 133	C_24_H_32_O_15_	R	[28]
	13	Methylsyringinoside	5.22	547.2039	548.2100	1.26	219, 191, 176, 161, 121, 93, 71	C_24_H_36_O_14_	R	[14]
	14	Barlerin	5.37	447.1514	448.1575	1.97	269, 161, 113, 101, 71	C_19_H_28_O_12_	S, R	-
	15	Dihydroferulic acid 4-*O*-glucuronide	5.78	371.0990	372.1051	1.67	179, 163, 121, 73	C_16_H_20_O_10_	S	[29]
	16	Yunnaneic acid D	5.79	539.1206	540.1262	1.82	297, 271, 197, 179, 161, 135, 109, 73	C_27_H_24_O_12_	S	[30]
	17	Lithospermic acid	5.81	537.1050	538.1106	2.11	339, 295, 269, 197, 179, 161, 135, 109, 73	C_27_H_22_O_12_	S	[24]
	18	Isofraxidin	5.98	221.0457	222.0523	1.30	177, 161, 145, 133, 123, 108, 95, 85, 67	C_11_H_10_O_5_	R	-
	19	Anchusoside-9	6.07	827.4449	828.4502	1.8	665, 503, 161, 113, 85, 71	C_42_H_68_O_16_	R	[31]
	20	Bayogenin triglycoside	6.09	1001.4954	1002.5030	0.17	942, 797, 635	C_49_H_78_O_21_	S, R	-
	21	Rosmarinic acid glucoside	6.14	521.1311	522.1368	2.13	359, 197, 179, 161, 135, 123, 73	C_24_H_26_O_13_	R	[32]
	22	Acetylanchusoside-9	6.22	869.4543	870.4608	0.33	707, 503, 161, 113, 85, 71	C_44_H_70_O_17_	R	[31]
	23	SA derivative I	6.33	537.1049	538.1106	2.73	285, 185, 135, 109, 121	C_27_H_22_O_12_	R	-
	24	Methylsyringin	6.40	385.1509	386.1571	1.57	207, 191, 176, 161, 121, 93, 71	C_18_H_26_O_9_	S, R	[14]
	25	Bayogenin diglycoside	6.42	839.4435	840.4502	1.00	633, 423, 161, 113, 85, 71	C_43_H_68_O_16_	S	[33]
	26	Salvianolic acid (SA) A	6.48	493.1150	494.1207	2.11	295, 267, 197, 185, 169, 135, 109, 73	C_26_H_22_O_10_	R	[34]
	27	Dihydroxybayogenin diglycoside	6.49	843.4406	844.4451	2.58	621, 459, 161, 113, 101, 71	C_42_H_68_O_17_	S	[35]
	28	SA derivative II	6.51	537.1046	538.1106	3.01	295, 185, 135, 109, 121	C_27_H_22_O_12_	R	-
	29	Rosmarinic acid (RA)	6.53	359. 0779	360.0840	1.95	197, 179, 161, 135, 123, 73, 62	C_18_H_16_O_8_	S, R	[36]
	30	Salvianolic acid (SA) E	6.70	717.1478	718.1528	1.72	339, 321, 295, 185, 161, 135, 109, 73	C_36_H_30_O_16_	R	[24]
	31	6″-Acetyl-methyl syringin	6.75	427.1616	428.1677	0.59	384, 219, 208, 191, 176, 161, 121, 93, 73	C_20_H_28_O_10_	R	-
	32	Clinopodic acid A	6.98	343.0829	344.0891	0.59	197, 179, 145, 135, 123, 117, 89, 73	C_18_H_16_O_7_	R	-
	33	Dehydro SA B	7.10	715.1324	716.1372	2.70	339, 295, 185, 135, 109, 72	C_36_H_28_O_16_	R	[24]
	34	Dehydro RA	7.00	357.0622	358.0683	0.70	197, 179, 161, 133, 123, 73	C_18_H_14_O_8_	R	[14]
	35	Methyl RA	7.06	373.0935	374.0996	0.65	197, 179, 161, 135, 123, 73	C_19_H_18_O_8_	S	[36]
	36	Citrinin	7.75	249.0771	250.0836	0.22	205, 157, 143, 122, 104	C_13_H_14_O_5_	R	[22]
	37	Malonylanchusoside-2	8.15	1027.5135	1028.5187	1.38	779, 659, 617, 599, 455, 159, 129, 113, 101, 87	C_51_H_80_O_21_	S	[37]
	38	Hydroxy Malonyl anchusoside-7	8.30	1043.5081	1044.5136	2.21	795, 659, 617, 471, 159, 129, 113, 101, 87	C_51_H_80_O_22_	S	-
	39	Hydroxy Dimalonylanchusoside 2/7	8.43	1129.5087	1130.5140	2.24	659, 471, 455, 159, 111, 101, 87	C_54_H_82_O_25_	S	-
	40	Malonylanchusoside-7	9.26	1027.5138	1028.5187	1.86	779, 659, 617, 599, 455, 161, 113, 101, 89	C_51_H_80_O_21_	S	[38]
	41	Gingerol	9.54	293.1662	294.1826	1.39	236, 221, 148, 127, 97, 72	C_17_H_26_O_4_	S, R	[22]
	42	Embellin	10.07	293.1766	294.1826	2.51	249, 193, 177, 136, 97, 79	C_17_H_26_O_4_	S, R	-

R_t_ = Retention time; Δ*m* = mass errors; [M-H]^−^ = *m*/*z* of the pseudomolecular ion in negative and positive ionization modes, respectively; *m*/*z* calcd = theoretical *m*/*z* value; R = roots; S = shoots; PMs = primary metabolites; SMs = secondary metabolites.

**Table 3 metabolites-12-00573-t003:** Chemical structure and fragmentation pattern of compound **19** and potential new compounds **22**, **31**, **37**, **40** identified by ESI-HRMS and MS/MS analysis.

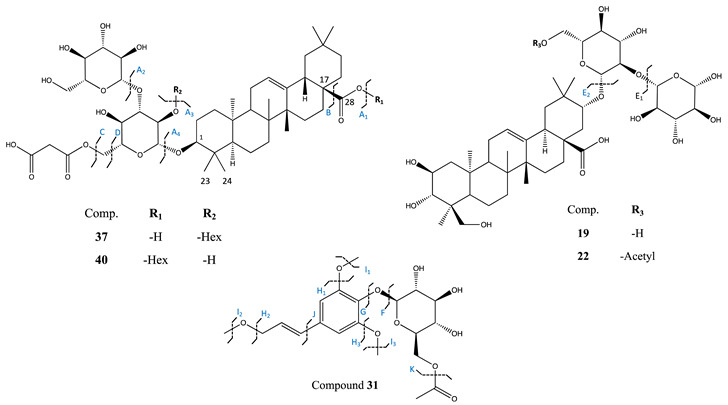		
Comp.	[M-H]^−^	MS/MS (*m*/*z*)
19	827.4449	665 (E_1_), 503 (E_2_), 161, 113, 85, 71
22	869.4543	707 (E_1_), 503 (E_2_), 161, 113, 85, 71
31	427.1616	384 (K), 219 (F), 208 (G), 191(G/I), 176 (G/H), 161 (G/H/I), 121(G/I/J), 93, 73
37	1027.5135	779 (A/C), 659 (B/A_1_/A_2_), 617 (A_1_/A_2_/C), 599 (A_1_/A_2_/D), 455 (A_4_), 159, 129, 113, 101, 87
40	1027.5138	779 (A/C), 659 (B/A_2_/A_3_), 617 (A_2_/A_3_/C), 599 (A_2_/A_3_/D), 455 (A_4_), 159, 129, 113, 101, 87

## Data Availability

Data is contained within the article or supplementary material.

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
