# Peer review of "The Metabolic Profile of Anchusa officinalis L. Differs According to Its Associated Arbuscular Mycorrhizal Fungi"

_metabolites, 2022, doi:10.3390/metabo12070573_

Round 1
Reviewer 1 Report
The subject treated in this manuscript is very interesting, although not entirely original. The manuscript presents a methodological error since there is no control. The authors should have the metabolome of the plant without mycorrhizal interaction, the metabolome of the mycorrhiza, and the metabolome of the plant mycorrhizal interaction. This way they will be able to carry out a discussion that accounts for the interaction that they want to show.
It would be advisable to indicate in table 2, the meaning of S and R.
Author Response
Reviewer 1:
- The subject treated in this manuscript is very interesting, although not entirely original. The manuscript presents a methodological error since there is no control. The authors should have the metabolome of the plant without mycorrhizal interaction, the metabolome of the mycorrhiza, and the metabolome of the plant mycorrhizal interaction. This way they will be able to carry out a discussion that accounts for the interaction that they want to show.
REPLY: We thank the reviewer for this comment. We agree that including a non-mycorrhized treatment in the study would have rendered the discussion part more solid and comprehensive. However, using the ‘’classical way’’ to compare samples which includes a direct comparison with control plants, would have rendered our discussion repetitive since this study represent the continuation of our precedent published experiment describing metabolome and morphological characteristic variations of Anchusa officinalis associated with Rhizophagus irregularis (https://doi.org/10.3389/fpls.2021.724352). In that analysis, non-mycorrhized plants have been used to evaluate the impact of the AMF-plant association. Indeed, specific metabolites were shown to be induced exclusively in the mycorrhized plants as compared to the non-mycorrhized plants (e.g., mono-, di- and trimers of caffeic acid and saponins derivatives). In continuation of that study, we investigated the impact on Anchusa officinalis in relation to different AMF strains belonging to the same genus (including Rhizophagus irregularis, extensively treated in our previous study). Different AMF-plant associations were directly compared to each other to identify, exclusively the expression of AMF species specific primary and secondary metabolites, significantly different in at least one combination AMF-plant associations. Thus, we rendered our study more targeted and focused. In fact, we wanted to investigate if the response remains the same when plants of A. officinalis are colonized with different AMF species belonging to the same genus. In the case that no differences between treatments were present, the PCA analysis would have resulted in a single cluster and the Volcano plots’ p-values and fold changes would have stayed below the threshold of significance. Therefore, no effect would have occurred in the metabolites production of A. officinalis plant treated with different AMF species.
In addition, we used the same plant species trough the experiment growing in the same environmental conditions, therefore, the observed differences could have been attributed only to the different AMF treatments. Indeed, our analysis evidenced differently induced compounds, statistically significant, related to specific plant-AMF symbiosis. This was exactly the point of our analysis. Moreover, as stated in the manuscript in lines 267-280, in nature A. officinalis plants are found associated with AMF and considering non-mycorrhized plants as control does not truly reflect the natural conditions. Rhizophagus irregularis was chosen as the reference strain for this study because it is considered as a model organism in AMF research. In this study, we only focused on the metabolome changes detected between the four AMF strains and we shed light on the different potential of each treatment in affecting A. officinalis’ biosynthetic pathways.
- It would be advisable to indicate in table 2, the meaning of S and R.
REPLY: We thank the reviewer for this remark. We added the meaning of S (Shoots) and R (Roots) in the legend of Table 2.
Reviewer 2 Report
The authors present a reasonable manuscript, containing novel observations and analyses that should be made available to the scientific community. There are, however, many points of criticism that need to be taken into account prior to bublication. The following remarks are meant to improve readability and, thus, impact of this contribution.
* The reviewer recommends to improve language throughout the text. Large parts of the manscript need to be re-written.
- Many things can be written easier and, thus, a lot clearer.
- Please avoid unnecessary wording. Example (there are many more): Instead of "... metabolome showed to be strongly modulated ..." write better: '... metabolome is strongly modulated ...'. Maybe "strongly" should also be avoided; this depends on comparing these results with those from other plants. Or: "These plants, as all plants, ..." should read: 'Plants interact with beneficial organisms.'.
- Please checke for grammar errors and similar.
- "... benificial organisms, often resulting in ("an" should be deleted) improved growth and health..." is trivial. That is exactly what "benificial" means. Why not simply (or similar): 'The interactions between plants and beneficial organisms improve not only health and growth, but modify the metabolonme considerably.'?
- Example: "... when associated to ..." should read '... associated with ...'.
- Example: "same genus AMF species ..." is not English.
* Please avoid abbreviations whereever possible. Abbreviations generally render texts hard to read. There is absolutely no need to abbreviate primary and secondary metabolites. Abstracts should in any case be written free from abbreviations, maybe with the exception of very common ones. To the very least, in case the authors want to stay with S-H, the meaning of the abbreviation needs to be repeated always, when first used in a paragraph. Please check the text for other abbreviations.
* Sometimes the authors can be much clearer without additional effort. Why, as an example, do they not use the correct 'phospate' instead of "phosphorus", which does not exist in normal soils and is certainly not a nutrient. The reviewer knows that the authors share word use with many in the community. Common use should never be the main argument, if aiming at clarity.
* Ideally, the paragraphs of a publication should be readable as stand-alone versions. There are many colleagues that don't read introductions, but are highly interested in novel results. It is not clever and does not improve readability and impact, if the Results section starts with "... was estimated on (the) plants, associated with the four AMF ...". Readability is considerably improved, if the authors mention the fungi at this place.
* Genus and species names must be given in italics throughout the text.
* The authors must check the quality of all figures with respect to readability. Symbols as well as inscriptions need to be readable also in printouts. In this respect all figures must be improved. The authors should - but must not necessarily - also consider, if they want to use the colours green and red for different things in the same figure. There are many among us, who cannot distinguish between them.
* Figure and Table captions must be understandable without referring to the main text. They should also allow basic interpretation of data. The authors must think again about abbreviations and those experimental details that are inevitable for this purpose.
* A general remark: In a scientific publication everything must be interesting, an , indeed, the authors describe important results in their manuscript. It is not advisable to start sentences too often with "Interestingly, ...". An example: Instead of writing "Interestingly, no differentiation was observed ..." the simple sentence 'No differentiation was observed ....' is much better.
* Skip 'tentative' in 2.2., especially in the title. According to the reviewer's opinion, the results are sound and important. Of course, they are not necessarily complete, but they are much more than "tentative".
* If possible, the authors should avoid remarks that sound like 'saving the world'. This is normally not appropriate for a fundamental scientific report. An example: "... symbiosis on earth that helps feed the world." Why not simply end the sentence after "symbiosis"? There are more such instances that make readers smile.
* Especially the discussion needs stringent restriction to important results and must also focus stronger on comparing these results with other studies on plant/fungus interactions. At the present stage, the discussion is lengthy and looks too much like a paragraph from a Master thesis or similar. The authors should re-write this part and must aim for clarity.
* Please check again throughout the text for meeting the journal's formal requirements.
* Check especially the Abstract and the Conclusions paragraphs for readability as stand-alone versions. Many read exclusively these parts, and abstracts are published in databases without the possibility to look also at other parts. The reviewer discourages the authors from using cryptic remarks in "Conclusions". Considering that this paragraph must be understandable on its own, the reviewer recommends not to use abbreviations and y compound numbers that cannot be immediately checked at this point.
The reviewer will be glad to promote this contribution with many novel observations after very careful revision and re-writing.
Author Response
Reviewer 2:
The authors present a reasonable manuscript, containing novel observations and analyses that should be made available to the scientific community. There are, however, many points of criticism that need to be taken into account prior to publication. The following remarks are meant to improve readability and, thus, impact of this contribution.
- The reviewer recommends to improve language throughout the text. Large parts of the manuscript need to be re-written.
REPLY: We thank the reviewer for the advices and comments concerning the grammar of the manuscript. We simplified sentences throughout the text and reviewed for grammatical errors (please see track changes in the manuscript). Furthermore, the paper has been carefully revised by colleagues with professional language editing skills to improve the grammar and readability.
- Please avoid abbreviations wherever possible. Abbreviations generally render texts hard to read. There is absolutely no need to abbreviate primary and secondary metabolites. Abstracts should in any case be written free from abbreviations, maybe with the exception of very common ones. To the very least, in case the authors want to stay with S-H, the meaning of the abbreviation needs to be repeated always, when first used in a paragraph. Please check the text for other abbreviations.
REPLY: We thank the reviewer for pointing this out. We removed any inappropriate abbreviation from the manuscript (e.g., primary metabolites, secondary metabolites and semi-hydroponic system). We tried to simplify as much as possible the readability of the manuscript by reducing considerably the number of abbreviations.
- Sometimes the authors can be much clearer without additional effort. Why, as an example, do they not use the correct 'phospate' instead of "phosphorus", which does not exist in normal soils and is certainly not a nutrient. The reviewer knows that the authors share word use with many in the community. Common use should never be the main argument, if aiming at clarity.
REPLY: We thank the reviewer for this remark. We changed the term phosphorus to phosphate.
- Ideally, the paragraphs of a publication should be readable as stand-alone versions. There are many colleagues that don't read introductions, but are highly interested in novel results. It is not clever and does not improve readability and impact, if the Results section starts with "... was estimated on (the) plants, associated with the four AMF ...". Readability is considerably improved, if the authors mention the fungi at this place.
REPLY: As per your request, we reconsidered each paragraph as a stand-alone version. To this purpose, we improved the readability of each paragraph by avoiding any abbreviation including the name of the studied AMF species.
- Genus and species names must be given in italics throughout the text.
REPLY: We thank the reviewer for the remark. We carefully reviewed the manuscript similar typing mistakes.
- The authors must check the quality of all figures with respect to readability. Symbols as well as inscriptions need to be readable also in printouts. In this respect all figures must be improved. The authors should - but must not necessarily - also consider, if they want to use the colours green and red for different things in the same figure. There are many among us, who cannot distinguish between them.
REPLY: We thank the reviewer for this comment. We improved the readability of the symbols (e.g., arrows) and the inscriptions of the Figures. We also changed the red and green colors in the Volcano-plots. However, we keep the colors in the PCA and the t-bars Figures because each color represents a specific treatment. We will ask the editor if it is possible to display the figures in a bigger format, since the images are now provided with the required quality criteria.
- Figure and Table captions must be understandable without referring to the main text. They should also allow basic interpretation of data. The authors must think again about abbreviations and those experimental details that are inevitable for this purpose.
REPLY: We thank the reviewer for this suggestion. We removed any abbreviation from the legends and add some extra information (e.g., AMF species name) to allow the basic interpretation of the data.
- A general remark: In a scientific publication everything must be interesting, an , indeed, the authors describe important results in their manuscript. It is not advisable to start sentences too often with "Interestingly, ...". An example: Instead of writing "Interestingly, no differentiation was observed ..." the simple sentence 'No differentiation was observed ....' is much better.
REPLY: We thank the reviewer for this advice. We removed the word “interestingly” from the manuscript to obtain simple sentences. The same has been applied in other parts of the manuscript.
- Skip 'tentative' in 2.2., especially in the title. According to the reviewer's opinion, the results are sound and important. Of course, they are not necessarily complete, but they are much more than "tentative".
REPLY: We have corrected this point. We removed the term tentative in the title of the section 2.2 and from the paragraph as well.
- If possible, the authors should avoid remarks that sound like 'saving the world'. This is normally not appropriate for a fundamental scientific report. An example: "... symbiosis on earth that helps feed the world." Why not simply end the sentence after "symbiosis"? There are more such instances that make readers smile
REPLY: Thank you for bringing this inconsistency to our attention. We have corrected this point and revised the manuscript for similar remarks (please see track changes in the manuscript).
- Especially the discussion needs stringent restriction to important results and must also focus stronger on comparing these results with other studies on plant/fungus interactions. At the present stage, the discussion is lengthy and looks too much like a paragraph from a Master thesis or similar. The authors should re-write this part and must aim for clarity.
REPLY: Thank you for pointing this out. We have revised the discussion focusing on most relevant results. Unfortunately, a limited number of bibliographic data related to most AMF species of this study are available. Most of them are already included in the discussion. Therefore, the implementation towards that direction cannot be much. However, we tried to focus stronger on comparing these results with other studies. We hope additional information’s in the manuscript will satisfy reviewer’s point of view.
- Please check again throughout the text for meeting the journal's formal requirements.
REPLY: We checked on the website, and we comply with all the requirements
- Check especially the Abstract and the Conclusions paragraphs for readability as stand-alone versions. Many read exclusively these parts, and abstracts are published in databases without the possibility to look also at other parts. The reviewer discourages the authors from using cryptic remarks in "Conclusions". Considering that this paragraph must be understandable on its own, the reviewer recommends not to use abbreviations and y compound numbers that cannot be immediately checked at this point.
REPLY: We thank the reviewer for this advice. We reconsidered the abstract and the conclusion parts, we removed any abbreviations and cryptic remarks. Furthermore, we removed the numbers of compounds and we included few examples as the reviewer suggested.
Round 2
Reviewer 1 Report
The response to my comment from the authors is not entirely satisfactory, but the manuscript can be published despite this. In these studies, it is always good to contrast the metabolome of all the individuals involved. Plants react to stress or interactions with other organisms by changing their metabolome. But it is also observed that compounds are transferred from the metabolome of the species with which they interact.